# Assessment of the effectiveness of public art in improving knowledge, attitude, practices and mitigation of stigmatization regarding neglected tropical diseases in South Eastern, Nigeria

Uchechukwu M. Chukwuocha[1]*, Ayoola O. Bosede[2], Chidubem D. Osuji[1], Adanna N. Chukwuocha[2], Harriet Abugewa[1], Amarachi Amawuru[1], Adesua Okoroh[1], Akuchi Echefula[1], Precious Nwabueze[1], Nathan Ukwenga[1], Alfred Barile[1]

1 Department of Public Health, Federal University of Technology, Owerri, Nigeria, 2 Department of Environmental Health Sciences, Federal University of Technology, Owerri, Nigeria

* uchukwuocha@gmail.com

## Abstract

### Background

Neglected Tropical Diseases (NTDs) continue to significantly impact marginalized communities, contributing to high morbidity, stigma, and social exclusion. Traditional health education strategies often fail to engage affected populations effectively. This study evaluates the impact of a community public art as an innovative tool for improving knowledge, shifting attitudes, enhancing preventive practices, and reducing stigma related to NTDs in Okwelle Community, Imo State, Nigeria.

### Methodology/Principal findings

A mixed-methods implementation research design was employed, involving 724 participants (362 at pretest and 362 at posttest). Public art installations depicting common NTDs in the study location including, Onchocerciasis, Buruli ulcer, and Lymphatic filariasis, were strategically placed in community spaces. Pretest and posttest surveys, key informant interviews, and focus group discussions were conducted to elucidate information about the influence of the public art on knowledge improvement and perception shifting. Quantitative data were analyzed using chi-square tests. Findings revealed a significant increase in NTDs awareness post-intervention ($p < 0.05$). Identification of disease symptoms improved, misconceptions regarding supernatural causation decreased, and preventive behaviors such as healthcare-seeking and participation in community health programs increased. Stigma reduction was evident, with greater acceptance of affected individuals and increased willingness to interact with them. Notably, 98.3% of participants supported maintaining public art as an educational tool.

**Data availability statement:** All relevant data are within the manuscript and its Supporting Information files

**Funding:** The author(s) received no specific funding for this work.

**Competing interests:** The authors have declared that no competing interests exist

## Conclusions/Significance

Public art can be a powerful and culturally relevant medium for enhancing NTDs awareness, promoting behavior change, and reducing stigma in endemic communities. This study highlights its potential for integration into broader public health strategies to improve disease awareness and community participation. Future research should explore the scalability and long-term sustainability of public art interventions in diverse settings.

### Author summary

Neglected Tropical Diseases affect some of the world's most vulnerable communities, causing illness, disability, and social stigma. Many people are unaware of these diseases or hold misconceptions that prevent early diagnosis and treatment. In this study, we explored how public art can be used as a tool to educate communities, improve knowledge, and reduce stigma associated with NTDs. We worked with community members in Okwelle, Nigeria, to create public murals that visually represented common NTDs including, Onchocerciasis, Buruli ulcer, and Lymphatic filariasis. Before the artwork was introduced, many residents had limited knowledge of these diseases, and some believed they were caused by witchcraft or punishment from a higher power. After the murals were displayed, awareness increased significantly, attitudes toward affected individuals improved, and more people took steps to prevent and treat NTDs.Our study shows that public art can be an effective and culturally relevant way to share health information, challenge stigma, and encourage healthier behaviors in underserved communities. This approach could be adapted for other public health challenges, making education more engaging and accessible to those who need it most.

## Introduction

Neglected Tropical Diseases (NTDs) are a collection of chronic, debilitating infections that affect over one billion of the world's poorest individuals, primarily in rural areas with limited access to healthcare, clean water, and adequate sanitation. Concentrated in under-resourced regions across Africa, Asia, and Latin America, NTDs result in over 200,000 deaths annually and cause about 19 million disability-adjusted life years (DALYs), with estimates equating the collective burden of NTDs to that of HIV/AIDS, tuberculosis, and malaria combined [1]. In rural communities, where economic opportunities are scarce, the disabling effects of NTDs trap affected individuals in a cycle of poverty, reducing productivity and increasing dependency [2,3]. This creates an immense socioeconomic impact that hinders not only individual quality of life but also broader community development.

Beyond the physical and economic toll, NTDs carry a significant social burden. Individuals affected by NTDs often face stigmatization and discrimination due to

the visible disfigurements and disabilities associated with these diseases [3]. This stigma manifests as social exclusion, shame, and a reluctance to seek timely medical care. Conditions like lymphatic filariasis, leprosy, and Buruli ulcer, which cause noticeable physical signs, are frequently misunderstood by community members, leading to increased stigmatization and isolation [3]. Such social ostracization exacerbates the challenges already faced by affected individuals, negatively impacting their mental well-being and further deepening health inequities. Moreover, stigma poses a major barrier to effective disease control efforts, as individuals may avoid diagnosis and treatment to escape judgment, inadvertently increasing disease transmission rates and impeding public health interventions [4].

To break the stigma surrounding NTDs, raising awareness and fostering positive perceptions within communities is crucial. Public health education initiatives can help demystify these diseases, addressing misconceptions and promoting empathy toward affected individuals [5,6]. However, traditional awareness campaigns often fail to engage rural populations effectively due to limited outreach, literacy barriers, and cultural disconnect [7]. Community-centered approaches that incorporate locally relevant and accessible methods are needed to ensure effective knowledge dissemination and engagement. One such approach is the use of public art, a creative and culturally resonant medium that can capture public attention, invoke shared experiences, and promote collective understanding. Public art, created for and by the community, holds a unique capacity to engage audiences, communicate health messages, and influence public perceptions [8,9]. Unlike conventional awareness materials, such as pamphlets or radio broadcasts, public art fosters interaction and dialogue, making health education a more immersive and participatory experience. By integrating key health messages into murals, sculptures, and other visual installations, public art can bridge communication gaps and encourage meaningful conversations about NTDs within affected communities [10]. Furthermore, art can provide a sense of ownership and agency among community members, empowering them to take an active role in health advocacy and stigma reduction efforts [11].

This study explored the potential of community public art under the initiative "Public Art for Public Health" as an innovative tool for promoting NTD awareness, enhancing knowledge, and reducing stigmatization in a rural community in Imo State, Nigeria. By assessing community knowledge, perceptions, and stigma levels both before and after the deployment of awareness-raising artwork, this research demonstrated the effectiveness of public art in shifting public attitudes and fostering social inclusion. The study also highlighted how art-driven interventions can complement existing health programs, contributing to broader NTDs control efforts by addressing not only the medical but also the social determinants of these diseases.

## Materials and methods

### Ethics Statement

This study received ethical approval from the Health Research Ethics Committee of the Department of Public Health, Federal University of Technology, Owerri, Nigeria with approval number 00010/FUTO/PUHEC/P116 before data collection began. All participants were fully informed about the purpose, procedures, and potential benefits of the study, and written informed consent was obtained from each participant prior to their involvement. Participation in the study was voluntary, and participants had the right to withdraw at any stage without any consequence. To ensure privacy and confidentiality, all personal identifiers were removed and data collection instruments anonymized. Data were securely stored and accessible only to the research team. Results were reported in aggregate form, with no individual identifiers disclosed, thereby safeguarding participant privacy. These measures adhered to ethical standards for research involving human participants, ensuring the protection of participants' rights and well-being throughout the study. The study was conducted in accordance with the principles of the Declaration of Helsinki.

### Research design

This study employed a quasi-experimental design with a one-group pretest-posttest approach [12], complemented by a mixed-methods strategy to assess the impact of the public art intervention on knowledge, attitude, practices and stigma related to NTDs

in Okwelle Community, Onuimo Local Government Area of Imo State, Nigeria. The study was part of the Public Art for Public Health initiative, where a team of researchers used strategically placed public artwork to foster knowledge improvement, shift perceptions, and mitigate stigma associated with NTDs.

The quasi-experimental one-group pretest-posttest design was selected due to practical and ethical considerations that made the inclusion of a control group unfeasible. This design allows for the measurement of changes in knowledge, attitudes, and practices (KAP) before and after the intervention, providing insights into its effectiveness. While it does not eliminate all threats to internal validity, it remains a widely accepted approach for assessing intervention outcomes when randomization or control groups are not feasible.

Also, a mixed-methods approach was employed to strengthen the study by integrating both quantitative and qualitative data. The quantitative analysis provided objective measures of changes in KAP, while the qualitative findings helped in contextualizing these changes by exploring participants' perceptions, experiences, and potential confounding factors [13,14].

## Study population

The study was carried out in the Okwelle community, Onuimo LGA, Imo State, Nigeria. This community has a population of approximately 9,247 (2024 projection of 2006 census) and is situated at latitude 5°45'5''N and longitude 7°10'39''E, with a landmass of approximately 8.5 km². Known for its central market, Okwelle attracts traders from surrounding towns in southeastern Nigeria, and the local economy is primarily based on agriculture, trade, and civil service employment.

Okwelle community was chosen as the study site due to its epidemiological relevance, accessibility, and strong community engagement potential. Data from the Imo State Ministry of Health (NTDs Office) indicate that the community bears a significant burden of neglected tropical diseases (NTDs) and related public health challenges, making it an appropriate setting for assessing the impact of the intervention. Also, anecdotal report reveals high level of stigmatization stemming from myths and misconceptions regarding NTDs among the people. The community has only one Basic Health Center that attend mostly to maternal and child health cases. A number of Lymphatic filariasis, Buruli ulcer and Onchocerciasis cases were sighted during data collection, confirming the prevalence of NTDs in the community.

The study population included all adults (18 years and above) residing within the Okwelle community for at least one year prior to the study.

## Sample size

The sample size for quantitative data collection was calculated using the Leslie Kish formula [15], where the estimated proportion of the target population (p) was based on similar studies [16]. With a prevalence estimate (p) of 63.1% from prior studies, the minimum sample size was calculated as 357.6, rounded to 362 to account for a 10% non-response rate.

## Sampling technique

For the cross-sectional study, the community was stratified into 26 streets, each assigned a unique number for easy identification. Systematic random sampling was then employed to select 361 households from the 26 streets giving an average of 14 households per street. The first household on each street was chosen randomly, followed by every sixth household. If a selected household lacked a suitable respondent or if the respondent was not interested in participating, the next eligible household was selected, while maintaining the interval of every sixth household until the 362-sample size was reached.

For the Key Informant Interviews (KIIs), which were conducted three months after the deployment of the public art as part of the follow-up study, a purposive sampling approach was used in collaboration with the basic health center. Key informants were identified and recruited based on their knowledge and influence on health-related matters within the community. Interviews continued until data saturation was achieved, resulting in a total of 16 key informants participating in the study.

## Inclusion and exclusion criteria

**Inclusion criteria.** Individuals aged 18 years and above who were residents of Okwelle community were eligible to participate in the study. Inclusion required willingness to provide informed consent and a minimum residency of one year to ensure familiarity with the local context. Participants needed to effectively communicate in English, Pidgin English, or the local dialect (Igbo) for the interviewer-administered questionnaire and Key Informant Interviews (KIIs). These criteria ensured that participants had adequate knowledge of community dynamics and could provide meaningful responses, enhancing the study's reliability and relevance to the local population.

**Exclusion criteria.** Individuals below 18 years of age were excluded from the study. These individuals were excluded from participation due to ethical considerations and the requirement for independent informed consent. Also, the survey tools and interview guides were designed for adult comprehension and self-reporting.

Additionally, residents who had lived in the community for less than one year were not eligible, as familiarity with the local context was essential. Individuals with cognitive impairments or severe health conditions that could hinder meaningful participation were also excluded. Furthermore, participants who declined to provide informed consent were not included in the study. These exclusion criteria ensured that only participants capable of providing reliable and informed responses were included, thereby maintaining the study's validity and accuracy in assessing community knowledge, attitudes, and practices regarding NTDs.

## Instruments for data collection

**Quantitative data.** Quantitative data were collected using a structured questionnaire administered through the Open Data Kit (ODK) tool, which enabled efficient data gathering in resource-constrained settings. The interviewer-administered approach to data collection made the use of ODK more appropriate and cost-effective. The questionnaire included sections on demographic characteristics, knowledge and perceptions of NTDs, and attitudes toward individuals with NTDs. Closed-ended questions with multiple-choice options and Likert scale items were used to capture respondents' attitudes and perceptions. The knowledge assessment tool included several questions related to the three most common NTDs in the community, including Onchocerciasis, Buruli ulcer, and Lymphatic filariasis. Trained research assistants, including six undergraduate students and one MPH student from the Department of Public Health, Federal University of Technology, Owerri, Imo State, Nigeria, were responsible for conducting the quantitative data collection.

Prior to data collection, they underwent comprehensive training on ethical research practices, the use of Open Data Kit (ODK), and effective participant engagement. ODK facilitated seamless data capture by allowing offline storage in areas with poor connectivity, ensuring data could be securely uploaded once an internet connection was available. No major challenges were encountered, as the research assistants were well-trained and adequately prepared for the exercise.

**Qualitative data.** A semi-structured key informant interview guide was used to collect qualitative data from key informants. The guide included prompts that encouraged informants to freely discuss their perceived changes in knowledge, attitudes, and stigma associated with NTDs within the community. The guide contains three major themes that participants were made to speak around, 1) Awareness and knowledge of NTDs in the community, 2) Attitudes towards NTDs and 3) Practices towards NTDs prevention and control. The key informants provided insights on the suitability and placement of the public art, as well as their observations of community engagement and interaction with the artwork.

**Interview prompts included.**

- Awareness and Knowledge of NTDs: "How would you describe the community's understanding of NTDs before and after the intervention?"

- Attitudinal Change and Stigma Reduction: "Have you observed any changes in how people perceive and interact with individuals affected by NTDs?"

- Improved Practices for NTDs Prevention and Control: "Do you think there has been an improvement in the community's efforts to prevent and control NTDs since the intervention? If so, in what ways?

The principal investigator and another lead researcher facilitated the key informant interviews (KIIs). The KIIs were conducted in a suitable location within the community, voice-recorded, and lasted an average of 30 minutes.

**Validity and reliability of instrument.** To ensure validity, the questionnaire was reviewed by a subject expert and the study supervisor, focusing on face and content validity. A pilot test of the instrument was conducted in Umudi, an NTD-endemic community within the same LGA as Okwelle, using 36 respondents (10% of the estimated sample size). The reliability of the questionnaire was assessed using Cronbach's alpha, yielding a reliability coefficient of 0.78, which meets the acceptable threshold for high reliability. Feedback from the pilot test were used to refine the questionnaire before full-scale data collection.

**Study implementation.** The processes involved in the implementation of the study are described below.

**Advocacy visit and stakeholders' identification.** A crucial first step involved engaging with community leadership in a focus group discussion (FGD) to assess the suitability of the already sketched artwork within the local context and to secure a highly accessible display location for all community members. Formal inquiries were made at the community health center through the chief nurse to identify key stakeholders who influence health opinions in the community. These stakeholders included the Traditional Ruler, Town Union Chairman, Traditional Herbalists, Market Union Leaders, Youth Leader, Women Leader, Health Workers, and Religious Leaders. A purposive sampling strategy was employed to achieve variation across these stakeholder groups while ensuring gender, age, and role diversity. To mitigate research bias inherent in non-probabilistic sampling, only stakeholders who had resided in the community for at least seven years and demonstrated an understanding of the local health dynamics were selected. Twelve [12] designated stakeholders participated in the FGDs that led to the final sketch of the artwork and the identification of the location where the artwork was painted.

During the engagement sessions, stakeholders were introduced to the concept of using public art as a medium for NTDs awareness, specifically focusing on Buruli ulcer, Onchocerciasis, and Lymphatic filariasis. Discussions were conducted in the Igbo language to ensure full comprehension among participants. The team explained that NTDs are not hereditary or supernatural ailments but are linked to environmental conditions and pathogenic factors, translated in Igbo as: *'Oria ndia abughi ihe na-efe efe, kama, o bu ihe n'esita na nje, nke n'esita na obibi anyi na usoro ime ihe n'ulo anyi di iche iche'.*

Stakeholders actively engaged in discussions, asked clarifying questions, and provided insights. They contributed suggestions to refine the concept before finalizing and displaying the artwork. Their involvement extended to identifying a suitable location for the artwork within the community to ensure maximum visibility and engagement.

**Baseline data collection.** Baseline data collection was conducted to establish initial information on the community's knowledge, perceptions, and stigmatization regarding NTDs. At this stage, only quantitative data were collected using an interviewer-administered questionnaire built on ODK; no qualitative interviews were conducted. The data gathered included participants' awareness of NTDs transmission, prevention, and treatment options. The baseline findings helped identify knowledge gaps and common stigmatizing attitudes, which informed the design and objectives of the subsequent public art intervention. Key Informant Interviews (KIIs) were conducted only during the follow-up phase to gain deeper insights into the intervention's impact.

**Intervention: Public art design and display.** A local Artist was engaged to design and create public art installations that conveyed information about NTDs, their prevention, and available treatments. A search was conducted to identify talented and respected local Artists, followed by meetings to discuss the project and gauge their interest. A community workshop was held to introduce the project to the artist and residents. This was done to generate interest and support by highlighting the role of art in raising awareness about NTDs.

The selected Artist collaborated with researchers to conceptualize the installations, ensuring key messages and visuals effectively conveyed NTD-related information. Inspiration was drawn from local culture, traditions, and community values. The Artist was provided with relevant NTDs data and educational materials, while healthcare professionals reviewed the content for medical accuracy.

Once the design was finalized, the Artist created murals, allowing community members to observe, ask questions, and engage in discussions. Interactive workshops were held involving residents in painting and decorating, fostering a sense of ownership and involvement. The murals were produced and displayed in a prominent public space close to the community market square, a location being frequently visited by everyone in and around the community.

**Follow-up data collection.** A follow-up data collection was conducted three months after the installation of the public art to evaluate its influence on community knowledge, perceptions, and stigmatization regarding NTDs. This follow-up aimed to assess changes in the target population's understanding of NTDs transmission, prevention, and treatment, as well as shifts in attitudes toward individuals affected by NTDs. The follow-up survey used the same questionnaire as the baseline data collection to ensure consistency in measuring changes. Data on knowledge scores, perceptions, and stigmatization levels were collected and compared with baseline data to assess the effectiveness of the public art intervention.

It is at this follow-up stage that the KIIs were conducted to determine their perceived changes in knowledge, attitudes, and stigma associated with NTDs within the community following the deployment of the public art.

## Data analysis

**Quantitative data.** This study collected two sets of data. The first set comprised baseline data collected before the deployment of the public artwork, aimed at assessing community knowledge about NTDs—including modes of transmission, prevention strategies, treatment options, and management. The baseline data also gauged perceptions of NTDs and the level of associated stigma. Three months after the artwork installation, a follow-up data collection was conducted using the same instrument with minor additions. The addition includes information on how many respondents had engaged with the artwork, their opinions of it, and whether they wanted it to remain in the community.

All collected data were initially processed in Microsoft Excel for cleaning, sorting, and coding, then imported into SPSS [17] for analysis. In SPSS, demographic information was analyzed using descriptive statistics and presented as frequency distributions and percentages, with certain variables displayed in tables and charts.

The answers to knowledge questions were presented as the frequency of correct responses. These frequencies were then compared between the pre-test and post-test groups, and the Chi-square test was used to assess the statistical significance of the differences observed. Cramer's V analysis was employed to determine the effect size of changes in variable outcomes, providing a measure of association strength between categorical variables and assessing the impact of public art interventions on NTD-related knowledge, attitudes, and practices. Cramer's V values range from 0 to 1, where 0 indicates no association between variables, while values closer to 1 suggest a stronger relationship. Conventionally, effect sizes are interpreted as follows: values below 0.10 indicate a negligible effect, 0.10–0.29 represent a small effect, 0.30–0.49 signify a moderate effect, and 0.50 or higher suggest a strong effect. This analysis allowed for a more nuanced understanding of the extent to which the intervention influenced behavioral and perceptual changes within the community. Categorical variables were presented as frequencies and percentages, and continuous variables were summarized with means and standard deviations.

Perceptions of NTDs were analyzed using a Likert scale, with baseline and follow-up frequencies displayed side-by-side. Stigmatization was assessed in two ways;

1. Respondents with NTDs or relatives with NTDs were asked if they had experienced stigma, with responses presented in a frequency distribution table, comparing baseline and follow-up responses.

2. Respondents without NTDs were asked if they would stigmatize others with NTDs. Their responses were also presented side-by-side with baseline and follow-up data to assess any changes in attitudes following the intervention.

**Qualitative data.**  The transcripts from the KIIs captured all spoken words, including non-verbal cues and pauses, ensuring a comprehensive record of the discussions. Each interview was stored as separate documents for systematic organization. NVivo, was employed to efficiently manage, code, and analyze the data. A coding framework was developed to tag and classify key themes, concepts, and variables relevant to the research. This structured approach facilitated the identification of patterns and insights, enabling a deeper understanding of community perspectives on NTDs, public art engagement, and stigma reduction.

**Trustworthiness of qualitative findings.**  To ensure the credibility of findings, triangulation was employed by comparing findings from KIIs, FGDs, and survey data. Member checking was conducted with select participants to validate interpretations. Transferability was enhanced through detailed descriptions of the study setting, including LGA and community characteristics, this will allow readers to assess applicability to other contexts. Dependability was ensured by maintaining an audit trail of methodological decisions and using a standardized interview guide across all KIIs. Confirmability was reinforced through peer debriefing, where external researchers were employed to review coding processes to mitigate bias and strengthen the reliability of interpretations.

## Results

### Sociodemographic characteristics of the study group

There were two groups of 362 persons each studied at pretest and posttest. The sociodemographic characteristics of the study groups are presented in Table 1, and shows that the characteristics of both groups were quite similar. The average age of the participants was 45.1±16.5 years in the pretest. For the posttest group, the mean age was 44.8±12.3 years. Close to half at each group had secondary education level (pretest=49.9%, posttest=48.9%), while those that had tertiary education level were 107 (30.1%) for pretest group and 123 (34%) for posttest group. In terms of marital status, majority were married in both groups (pretest=66.2%; posttest=70.7%). The occupational status for the pretest group includes 47.3% self-employed, 27% employed and 25.6% unemployed. For the posttest we have self-employed (45%), employed (34.5%) and unemployed (20.4%). A total of 31.3% and 31.2% respectively in pretest and posttest were doing hand work, while 29.9% were involve in trading in pretest group (posttest=29%).

The less than thirty thousand naira (N30,000) monthly income earners make up 30.4% of the pretest group (11.3% in posttest group), while about 4.5% in the pretest earn at least a N120,000 in a month (20.2% in posttest). Those with household size of 1–3 persons in the pretest group were 148 (41.7%), while 191 (52.8%) of the same household size were in the posttest group.

### Knowledge of Onchocerciasis at pretest and posttest

Table 2 shows that awareness of Onchocerciasis increased from 28.5% at pretest to 39% at posttest, with a significant difference ($\chi^2$=4.409, p=0.036) though the effect size was small (V=0.078). Identification of Onchocerciasis manifestations also improved significantly from 41.7% to 53.0% ($\chi^2$=9.313, p=0.002), but the effect size remained small (V=0.113).

Understanding that NTDs can disproportionately affect certain groups rather than everyone equally improved significantly from 25.1% at pretest to 40.9% at posttest, while uncertainty about who is most affected by NTDs decreased from 71% to 51.7% ($\chi^2$=29.44, p<0.001). The effect size for this shift was moderate (V=0.202), indicating meaningful knowledge gains.

Awareness of associated problems increased, with disability (18% to 33%), loss of employment (19% to 23%), poor health (13.3% to 19.6%), and death (5.8% to 14.9%) all showing significant improvement. The proportion of respondents unaware of NTD-related problems declined from 40.3% to 8.8% ($\chi^2$=184.74, p<0.001), with a large effect size (V=0.505), reflecting a strong shift in understanding.

**Table 1. Sociodemographic characteristics of the study group.**

| Sociodemographic Factors | Pretest | | Posttest | |
|---|---|---|---|---|
| | Freq | Percent (%) | Freq | Percent (%) |
| Age (Mean±SD) | 45.10±16.47 | | 44.81±12.29 | |
| **Educational attainment** | | | | |
| Secondary school | 177 | 49.86 | 177 | 48.9 |
| Tertiary | 107 | 30.14 | 123 | 33.98 |
| Primary school | 52 | 14.65 | 39 | 10.77 |
| Others | 19 | 5.35 | 23 | 6.35 |
| **Marital Status** | | | | |
| Married | 235 | 66.2 | 256 | 70.72 |
| Single | 76 | 21.41 | 46 | 12.71 |
| Widowed | 33 | 9.3 | 30 | 8.29 |
| Divorced | 11 | 3.1 | 30 | 8.29 |
| **Occupational status** | | | | |
| Self Employed | 168 | 47.32 | 163 | 45.03 |
| Employed | 96 | 27.04 | 125 | 34.53 |
| Unemployed | 91 | 25.63 | 74 | 20.44 |
| Occupation | | | | |
| Hand work | 111 | 31.27 | 113 | 31.22 |
| Trading | 106 | 29.86 | 105 | 29.01 |
| Farming | 72 | 20.28 | 63 | 17.4 |
| White collar | 66 | 18.59 | 81 | 22.38 |
| **Income level** | | | | |
| N0- N30,000 | 108 | 30.42 | 41 | 11.33 |
| N31,000-N60,000 | 103 | 29.01 | 115 | 31.77 |
| N61,000-N90,000 | 81 | 22.82 | 49 | 13.54 |
| N90,000-N120,000 | 47 | 13.24 | 84 | 23.2 |
| N120,000 and above | 16 | 4.51 | 73 | 20.17 |
| **Number of people living in household** | | | | |
| 1-3persons | 148 | 41.69 | 191 | 52.76 |
| 4-6persons | 158 | 44.51 | 156 | 43.09 |
| 7 persons and above | 49 | 13.8 | 15 | 4.14 |

Knowledge of Onchocerciasis transmission via black fly bites increased from 40.1% to 54.7%, with a moderate effect size ($V = 0.249$). Understanding that Onchocerciasis is not transmitted from person-to-person also improved from 28.2% to 54.4% ($x^2 = 55.63$, $p < 0.001$), with a moderate effect size ($V = 0.277$).

Awareness of preventive measures increased, with responses for vaccination decreasing from 33.1% to 20.4%, and avoiding black fly bites increasing from 24.9% to 49.7% ($x^2 = 6.758$, $p = 0.030$), but the effect size was small ($V = 0.097$). Adoption of protective measures such as personal hygiene, wearing long sleeves, and using insect repellents also improved significantly ($x^2 = 48.65$, $p < 0.001$), with a moderate effect size ($V = 0.259$).

At pretest, 49.4% were unaware of treatment options for Onchocerciasis, but posttest findings showed that 61.9% recognized Ivermectin/Mectizan as a treatment ($x^2 = 28.96$, $p < 0.001$). The effect size for this improvement was moderate ($V = 0.200$), suggesting a notable increase in treatment awareness.

**Table 2. Knowledge of onchocerciasis at pretest and posttest.**

| Knowledge of Onchocerciasis | Pretest | | Posttest | | Chi-sq | V | P |
|---|---|---|---|---|---|---|---|
| | Freq | % | Freq | % | (χ²) | | |
| **Ever heard of Onchocerciasis** | | | | | | | |
| Yes | 143 | 39.5 | 171 | 47.2 | | | |
| No | 219 | 60.5 | 191 | 52.8 | 4.409 | 0.078 | 0.036 |
| **Identification of Onchocerciasis manifestation from the provided photo album** | | | | | | | |
| Correct | 151 | 41.7 | 192 | 53.0 | | | |
| Incorrect | 211 | 58.3 | 170 | 47.0 | 9.313 | 0.113 | 0.002 |
| **Groups most disproportionately affected by NTDs** | | | | | | | |
| Affects everyone equally | 91 | 25.1 | 148 | 40.9 | | | |
| Men | 9 | 2.5 | 14 | 3.9 | | | |
| Women | 4 | 1.1 | 9 | 2.5 | | | |
| Children | 1 | 0.3 | 4 | 1.1 | | | |
| Don't know | 257 | 71.0 | 187 | 51.7 | 29.440 | 0.202 | <0.001 |
| **Problems associated with NTDs** | | | | | | | |
| Disability | 65 | 18.0 | 120 | 33.1 | | | |
| Loss of employment | 69 | 19.1 | 83 | 22.9 | | | |
| Poor health | 48 | 13.3 | 71 | 19.6 | | | |
| Death | 21 | 5.8 | 54 | 14.9 | | | |
| Spending on healthcare and economic impoverishment etc | 13 | 3.6 | 32 | 8.8 | | | |
| Don't know | 146 | 40.3 | 2 | 0.6 | 184.74 | 0.505 | <0.001 |
| **Means of contracting Onchocerciasis** | | | | | | | |
| Poor hygiene | 65 | 18.0 | 89 | 24.6 | | | |
| Infection through bites of black flies | 145 | 40.1 | 198 | 54.7 | | | |
| Witchcraft | 60 | 16.6 | 22 | 6.1 | | | |
| Affliction by the gods | 32 | 8.8 | 9 | 2.5 | | | |
| Unknown | 60 | 16.6 | 44 | 12.2 | 44.91 | 0.249 | <0.001 |
| **Onchocerciasis transmiting from person to person in the community** | | | | | | | |
| I don't know | 186 | 51.4 | 133 | 36.7 | | | |
| No | 102 | 28.2 | 197 | 54.4 | | | |
| Yes | 74 | 20.4 | 32 | 8.8 | 55.63 | 0.277 | <0.001 |
| **Onchocerciasis prevention** | | | | | | | |
| I Don't know | 102 | 28.2 | 78 | 21.5 | | | |
| Vaccination | 120 | 33.1 | 74 | 20.4 | | | |
| Avoiding being biting by blackflies | 90 | 24.9 | 180 | 49.7 | | | |
| Appeasing the gods through sacrifice | 20 | 5.5 | 20 | 5.5 | | | |
| Cannot be prevented | 21 | 5.8 | 7 | 1.9 | | | |
| Spiritual fortification | 9 | 2.5 | 3 | 0.8 | 5.333 | 0.097 | 0.069 |
| **Personal protective measures of Onchocerciasis** | | | | | | | |
| Personal hygiene | 72 | 19.9 | 98 | 27.1 | | | |
| Wearing of long sleeves and trousers | 66 | 18.2 | 86 | 23.8 | | | |
| Wearing of insect repellant | 58 | 16.0 | 85 | 23.5 | | | |
| Wearing of permethrin-treated clothes | 52 | 14.4 | 39 | 10.8 | | | |
| Eating good food | 54 | 14.9 | 10 | 2.8 | | | |
| Community spiritual cleansing | 33 | 9.1 | 23 | 6.4 | | | |
| Appeasing the gods | 21 | 5.8 | 12 | 3.3 | | | |
| Going to bed early | 6 | 1.7 | 9 | 2.5 | 48.65 | 0.259 | <0.001 |

*(Continued)*

**Table 2.** (Continued)

| Knowledge of Onchocerciasis | Pretest | | Posttest | | Chi-sq | V | P |
|---|---|---|---|---|---|---|---|
| | Freq | % | Freq | % | (χ²) | | |
| **Treatments choices for onchocerciasis** | | | | | | | |
| Don't know | 179 | 49.4 | 109 | 30.1 | | | |
| Ivermectin/Mectizan | 157 | 43.4 | 224 | 61.9 | | | |
| Rifampicin | 19 | 5.2 | 21 | 5.8 | | | |
| Doxycycline | 7 | 1.9 | 8 | 2.2 | 28.96 | 0.200 | <0.001 |

## Knowledge of Buruli ulcer at pretest and posttest

Table 3 shows that awareness of Buruli ulcer increased from 53.6% at pretest to 75.7% at posttest ($\chi^2 = 40.186$, $p < 0.001$, V = 0.333), indicating a moderate effect size. Identification of Buruli ulcer manifestations from a photo album improved from 52.2% to 71.5% posttest ($\chi^2 = 5.905$, $p = 0.015$, V = 0.128), but with a small effect size.

**Table 3.** Knowledge of Buruli Ulcer at pretest and posttest.

| Knowledge of Neglected Tropical Diseases | Pretest | | Posttest | | (χ²) | V | p |
|---|---|---|---|---|---|---|---|
| | Freq | % | Freq | % | | | |
| **Knowledge of Buruli Ulcer** | | | | | | | |
| **Ever heard of Buruli ulcer** | | | | | | | |
| Yes | 194 | 53.6 | 274 | 75.7 | | | |
| No | 168 | 46.4 | 88 | 24.3 | 40.186 | 0.333 | <0.001 |
| **Identification of Buruli ulcer from provided photo album** | | | | | | | |
| Correct | 189 | 52.2 | 259 | 71.5 | | | |
| Incorrect | 5 | 1.4 | 22 | 6.1 | 5.905 | 0.128 | 0.015 |
| **Causes of Buruli ulcer** | | | | | | | |
| Poison | 178 | 49.2 | 43 | 11.9 | | | |
| Witch craft | 126 | 34.8 | 35 | 9.7 | | | |
| Bacteria | 97 | 26.8 | 176 | 48.6 | | | |
| Curse | 58 | 16.0 | 11 | 3.0 | | | |
| Insect bites | 21 | 5.8 | 108 | 29.8 | 237.77 | 0.405 | <0.001 |
| **Buruli ulcer is communicable** | | | | | | | |
| No | 102 | 28.2 | 224 | 61.9 | | | |
| Don't know | 191 | 52.8 | 99 | 27.3 | | | |
| Yes | 69 | 19.1 | 39 | 10.8 | 83.18 | 0.339 | <0.001 |
| **Personal measures to prevent Buruli ulcer** | | | | | | | |
| Promptly wash scratches or cuts with soap | 129 | 35.6 | 59 | 16.3 | | | |
| Use of topical antiseptic and dressing on wound | 116 | 32.0 | 64 | 17.7 | | | |
| Wearing insect repellent on any exposed skin | 122 | 33.7 | 206 | 56.9 | | | |
| BCG vaccination | 61 | 16.9 | 71 | 19.6 | | | |
| Wearing of gloves, long sleeved and trousers | 41 | 11.3 | 109 | 30.1 | 92.70 | 0.253 | <0.001 |
| **Safe and effective medication available for the treatment of Buruli ulcer** | | | | | | | |
| Don't know | 268 | 74.0 | 92 | 25.4 | | | |
| Rifampicin | 36 | 9.9 | 109 | 30.1 | | | |
| Streptomycin | 44 | 12.2 | 106 | 29.3 | | | |
| Clarithromycin | 14 | 3.9 | 96 | 26.5 | 207.95 | 0.438 | <0.001 |

Misconceptions about the disease also reduced. At pretest, 49.2% believed Buruli ulcer was caused by poison, while at posttest, 48.6% correctly identified bacteria as the cause ($\chi^2 = 237.77$, $p < 0.001$, $V = 0.405$), suggesting a moderate-to-large effect size. Knowledge that Buruli ulcer is not transmitted person-to-person increased significantly from 28.2% to 61.9% posttest ($\chi^2 = 83.18$, $p < 0.001$, $V = 0.339$), with a moderate effect size.

Awareness of personal protective measures improved, with 56.9% at posttest acknowledging the use of insect repellents on exposed skin compared to 33.7% at pretest ($\chi^2 = 92.70$, $p < 0.001$, $V = 0.253$), reflecting a small-to-moderate effect size. Similarly, 30.1% recognized the importance of wearing gloves, long sleeves, and trousers, up from 11.3% at pretest.

Regarding treatment knowledge, 74% at pretest were unaware of the availability of safe and effective medication for Buruli ulcer, which significantly decreased to 25.4% at posttest ($\chi^2 = 207.95$, $p < 0.001$, $V = 0.438$), showing a large effect size.

### Knowledge of Lymphatic filariasis at pretest and posttest

Table 4 indicates that awareness of Lymphatic filariasis increased from 57.7% (209 respondents) at pretest to 63.5%, but this change was not statistically significant ($\chi^2 = 2.552$, $p = 0.110$, $V = 0.053$), indicating only a small effect.

However, identification of Lymphatic filariasis manifestations from a photo album improved significantly from 33.1% at pretest to 61.3% at posttest ($\chi^2 = 57.657$, $p < 0.001$, $V = 0.277$), demonstrating a moderate effect size.

Understanding of disease causation also improved. At pretest, only 15.5% and 10.8% correctly identified mosquito bites and a dirty environment as causes, increasing to 37% and 29.8% at posttest. Meanwhile, misconceptions such as witchcraft (20.4% to 4.7%) and affliction (17.4% to 4.1%) significantly declined. The overall change in knowledge regarding causation was highly significant ($\chi^2 = 139.48$, $p < 0.001$) with a large effect size ($V = 0.440$), suggesting a substantial improvement in understanding.

The proportion of respondents unaware of the transmission mode dropped from 41.2% to 26.8%, while correct identification of mosquito bites as the mode of transmission increased from 24.9% to 48.6% ($\chi^2 = 57.98$, $p < 0.001$, $V = 0.277$), reflecting a moderate effect size.

Knowledge of preventive measures also showed substantial improvement. At posttest, 37.0% identified wearing protective clothing, up from 16.3% at pretest, while recognition of mosquito bite avoidance as a preventive measure increased from 22.1% to 35.4%. These changes were statistically significant ($\chi^2 = 89.98$, $p < 0.001$, $V = 0.348$), indicating a moderate-to-large effect size.

More than half (53%) of respondents were unaware of safe and effective medication for lymphatic filariasis at pretest, but this significantly decreased to 23.8% at posttest. Awareness of Albendazole/Ivermectin as a treatment increased from 16.3% to 27.1%, and Ivermectin alone from 13.0% to 24.9% ($\chi^2 = 67.80$, $p < 0.001$, $V = 0.309$), representing a moderate-to-large effect size.

At a 5% significance level, all posttest changes were highly significant ($p < 0.001$), except for general awareness of lymphatic filariasis, which did not show a statistically significant change. The largest knowledge gains were observed in identification of symptoms, understanding of disease causation, and awareness of transmission and prevention methods, with effect sizes ranging from moderate to large ($V = 0.277 – 0.440$), indicating a meaningful impact of the intervention.

### Attitudes towards NTDs in the study area

Table 5 indicates improved attitude towards NTDs at posttest, as many disagreed against wrong insinuations on the NTDs. At pretest, over 65% disagreed that NTDs are agelong diseases that can be inherited from generation to generation. This increased at posttest to 83.1%. Similarly, many also disagreed that "NTDs are repercussion for sinful act against God/Allah" at posttest (36.2% to 49.4%) and about three folds strongly disagreed at posttest that Most NTDs are infectious diseases which are contracted when people come in contact with the infectious agents (15.5% to 46.4%). Other items also showed reasonable improvement in opinions and attitude against NTDs from pretest to posttest. These include

**Table 4. Knowledge of lymphatic filariasis at pretest and posttest.**

| Knowledge of Lymphatic filariasis | Pretest | | Posttest | | (χ2) | V | P |
|---|---|---|---|---|---|---|---|
| | **Freq** | **%** | **Freq** | **%** | | | |
| **Ever heard of Lymphatic filariasis** | | | | | | | |
| Yes | 209 | 57.7 | 230 | 63.5 | | | |
| No | 153 | 42.3 | 132 | 36.5 | 2.552 | 0.053 | 0.110 |
| **Identification of Lymphatic filariasis manifestation from provided photo album** | | | | | | | |
| Correct | 120 | 33.1 | 222 | 61.3 | | | |
| Incorrect | 242 | 66.9 | 140 | 38.7 | 57.657 | 0.277 | <0.00! |
| **Causes of Lymphatic filariasis** | | | | | | | |
| Don't know | 105 | 29.0 | 69 | 19.1 | | | |
| Witcraft | 74 | 20.4 | 17 | 4.7 | | | |
| Affliction | 63 | 17.4 | 15 | 4.1 | | | |
| Mosquito bite | 56 | 15.5 | 134 | 37.0 | | | |
| Dirty environment | 39 | 10.8 | 108 | 29.8 | | | |
| Eating contaminated food | 23 | 6.4 | 19 | 5.2 | | | |
| Sexual intercourse | 2 | 0.6 | 0 | 0.0 | 139.48 | 0.440 | <0.001 |
| **Mode of transmission of Lymphatic filariasis** | | | | | | | |
| Don't know | 149 | 41.2 | 97 | 26.8 | | | |
| Affliction | 94 | 26.0 | 46 | 12.7 | | | |
| Mosquito bites | 90 | 24.9 | 176 | 48.6 | | | |
| Sharing contaminated clothing | 29 | 8.0 | 43 | 11.9 | 57.98 | 0.277 | <0.001 |
| **Prevention of Lymphatic filariasis** | | | | | | | |
| I don't know | 149 | 41.2 | 77 | 21.3 | | | |
| Wearing protective clothing | 59 | 16.3 | 134 | 37.0 | | | |
| Avoid mosquito bites | 80 | 22.1 | 128 | 35.4 | | | |
| Appeasing the gods | 39 | 10.8 | 12 | 3.3 | | | |
| Spiritual sacrifice | 35 | 9.7 | 11 | 3.0 | 89.98 | 0.348 | <0.001 |
| **Safe and effective medication for lymphatic filariasis?** | | | | | | | |
| Don't know | 192 | 53.0 | 86 | 23.8 | | | |
| Albendazole/Ivermectin | 59 | 16.3 | 98 | 27.1 | | | |
| Ivermectin alone | 47 | 13.0 | 90 | 24.9 | | | |
| Diethylcarbamazine citrate | 21 | 5.8 | 23 | 6.4 | | | |
| Albendazole alone | 11 | 3.0 | 23 | 6.4 | 67.80 | 0.309 | <0.001 |

NTDs is caused as a result of manipulation from witchcraft (strongly disagreed: 23.5% to 43.5%), NTDs only affect the poor (strongly disagreed: 30.9% to 45%) and NTDs cannot be prevented, it comes whenever it wants to come (strongly disagreed: 30.7% to 44.6%).

### Effect of public art in improving the practices of NTDs control

Table 6 shows that those who sought medical treatment or advice for NTD-related symptoms or conditions for themselves or a family member in the past year increased significantly from 26% at pretest to 59.1% at posttest (p = 0.004, χ² = 8.220, V = 0.151), indicating a small effect size.

Frequent engagement in practices that help prevent NTDs, such as personal hygiene, sanitation, or use of protective measures, increased from 25.1% to 46.4%, while those who never engaged in such practices reduced from 8.6% to 4.4% (p < 0.001, χ² = 36.836, V = 0.184), showing a small-to-moderate effect.

**Table 5. Attitude towards NTDs in the study Area.**

| Attitude | Pretest | | Posttest | |
|---|---|---|---|---|
| | **Freq** | **%** | **Freq** | **%** |
| **NTDs are agelong diseases that can be inherited from generation to generation** | | | | |
| Strongly disagree | 122 | 30.9 | 163 | 45.0 |
| Disagree | 114 | 34.3 | 138 | 38.1 |
| Agree | 92 | 25.4 | 42 | 11.6 |
| Strongly agree | 34 | 9.4 | 21 | 5.8 |
| **NTDs are repercussion for sinful act against God/Allah** | | | | |
| Strongly Disagree | 131 | 36.2 | 179 | 49.4 |
| Disagree | 104 | 28.7 | 109 | 30.1 |
| Agree | 92 | 25.4 | 48 | 13.3 |
| Strongly agree | 35 | 9.7 | 26 | 7.2 |
| **Most NTDs are infectious diseases which are contracted when people come in contact with the infectious agents** | | | | |
| Strongly disagree | 56 | 15.5 | 168 | 46.4 |
| Disagree | 116 | 32.0 | 108 | 29.8 |
| Agree | 141 | 39.0 | 62 | 17.1 |
| Strongly agree | 49 | 13.5 | 24 | 6.6 |
| **NTDs is caused as a result of manipulation from witchcraft** | | | | |
| Strongly disagree | 85 | 23.5 | 164 | 45.3 |
| Disagree | 75 | 20.7 | 106 | 29.3 |
| Agree | 163 | 45.0 | 72 | 19.9 |
| Strongly agree | 39 | 10.8 | 20 | 5.5 |
| **There is no treatment for NTDs** | | | | |
| Strongly Disagree | 102 | 28.2 | 173 | 47.8 |
| Disagree | 120 | 33.1 | 120 | 33.1 |
| Agree | 103 | 28.5 | 44 | 12.2 |
| Strongly agree | 37 | 10.2 | 25 | 6.9 |
| **NTDs only affects the poor** | | | | |
| Strongly disagree | 112 | 30.9 | 163 | 45.0 |
| Disagree | 120 | 33.1 | 127 | 35.1 |
| Agree | 105 | 29.0 | 53 | 14.6 |
| Strongly agree | 25 | 6.9 | 19 | 5.2 |
| **NTDs cannot be prevented, it comes whenever it wants to come** | | | | |
| Strongly Disagree | 111 | 30.7 | 165 | 44.6 |
| Disagree | 120 | 33.1 | 119 | 32.9 |
| Agree | 102 | 28.2 | 69 | 19.1 |
| Strongly agree | 29 | 8.0 | 19 | 5.2 |

Participation in community health programs or initiatives related to NTDs rose significantly from 47.8% at pretest to 71.3% at posttest (p<0.001, χ²=41.422, V=0.338), suggesting a moderate effect size.

The extent to which people actively shared information about NTDs and their prevention methods with friends and family improved significantly from 37.8% to 68.0% (p<0.001, χ²=65.86, V=0.427), demonstrating a strong effect.

Belief that NTD treatment drugs should be prescribed by a doctor before use was already high at baseline (73.5%) but increased further to 85.4% post-intervention (p<0.001, χ²=22.73, V=0.251), indicating a small-to-moderate effect.

**Table 6. Effect of public art in improving the control practices for NTDs among community members in study area (n = 362).**

| Practices for NTDs Control | Pretest | | Posttest | | (χ²) | V | P |
|---|---|---|---|---|---|---|---|
| | Freq | % | Freq | % | | | |
| **Ever sought medical treatment for NTD-related symptoms in the past year** | | | | | | | |
| Yes | 94 | 26 | 214 | 59.1 | | | |
| No | 268 | 74 | 148 | 40.9 | 8.220 | 0.151 | 0.004 |
| **Frequency of engaging in NTDs prevention practices** | | | | | | | |
| Frequently | 91 | 25.1 | 168 | 46.4 | | | |
| Occasionally | 157 | 43.4 | 121 | 33.4 | | | |
| Rarely | 82 | 22.7 | 57 | 15.7 | | | |
| Never | 31 | 8.6 | 16 | 4.4 | 36.836 | 0.184 | < 0.001 |
| **Participation in NTDs-related community programs** | | | | | | | |
| Yes | 173 | 47.8 | 258 | 71.3 | | | |
| No | 189 | 52.2 | 104 | 28.7 | 41.422 | 0.338 | < 0.001 |
| **Active sharing of NTDs-related information** | | | | | | | |
| Yes | 137 | 37.8 | 246 | 68.0 | | | |
| No | 225 | 62.2 | 116 | 32.0 | 65.86 | 0.427 | <0.001 |
| **NTDs medications should be prescribed by a doctor** | | | | | | | |
| TRUE | 266 | 73.5 | 303 | 85.4 | | | |
| FALSE | 96 | 26.5 | 52 | 14.6 | 22.73 | 0.251 | <0.001 |

Overall, statistical tests at a 5% significance level confirm that the introduction of public art had a significant impact on improving NTD control practices among community members in the study area (p < 0.05).

## Effect of public art in reducing social stigmatization surrounding NTDs

As shown in Table 7, the proportion of respondents who had personally experienced or knew someone affected by NTDs remained stable between pretest (26.8%) and posttest (28.5%) (χ² = 0.257, p = 0.612, V = 0.027), indicating no significant change in disease awareness. However, a downward trend was observed in specific stigma-related experiences. The proportion of those prevented from attending an assembly due to NTDs declined from 55.6% to 40.8% (χ² = 4.463, p = 0.035, V = 0.208), demonstrating a small effect size. Similarly, those denied entrance into public places due to NTDs showed a non-significant decrease from 52.6% to 46.6% (χ² = 0.719, p = 0.397, V = 0.083).

Other stigma-related experiences, such as handshake denial (χ² = 0.759, p = 0.384, V = 0.086), difficulties in personal relationships (χ² = 1.977, p = 0.160, V = 0.141), employment denial (χ² = 1.415, p = 0.234, V = 0.121), and social isolation (χ² = 2.651, p = 0.104, V = 0.166), showed non-significant reductions. Also, the proportion of respondents who reported children being excluded from school due to NTDs remained unchanged (χ² = 0.169, p = 0.681, V = 0.042).

In contrast, there were significant improvements in attitudes toward individuals affected by NTDs. The proportion of respondents willing to sit in the same car with an affected person increased from 27.9% to 42.0% (χ² = 22.07, p < 0.001, V = 0.247), indicating a small-to-moderate effect. Readiness to allow children to attend school where there is a known NTD case increased substantially from 22.5% to 43.1% (χ² = 45.42, p < 0.001, V = 0.354), reflecting a moderate effect. Similarly, the willingness to employ an individual with NTDs improved significantly from 14.9% to 32.3% (χ² = 36.20, p < 0.001, V = 0.317).

In addition, improvements were observed in interpersonal acceptance. The proportion of respondents willing to remain friends with someone who develops an NTD increased from 30.4% to 43.1% (χ² = 18.14, p < 0.001, V = 0.224). Acceptance of buying from a vendor affected by NTDs improved from 22.3% to 36.2% (χ² = 21.97, p < 0.001, V = 0.241). Furthermore,

Diseases

**Table 7. Effect of public art in reducing social stigmatization in the study area.**

| | Pretest | | Posttest | | $(x^2)$ | V | P |
|---|---|---|---|---|---|---|---|
| **Likely Stigmatization factors** | **Freq** | **%** | **Freq** | **%** | | | |
| Experience of the NTDs | | | | | | | |
| Yes | 95 | 26.8 | 103 | 28.5 | | | |
| No | 260 | 73.2 | 259 | 71.6 | 0.257 | 0.027 | 0.612 |
| Ever prevented from an assembly due to NTDs | n = 95 | | n = 103 | | | | |
| Yes | 53 | 55.6 | 42 | 40.8 | | | |
| No | 42 | 44.4 | 61 | 59.2 | 4.463 | 0.028 | 0.035 |
| Denied entrance into a place because of NTDs | | | | | | | |
| Yes | 50 | 52.6 | 48 | 46.6 | | | |
| No | 45 | 47.4 | 55 | 53.4 | 0.719 | 0.083 | 0.397 |
| Denied handshake due to NTDs | | | | | | | |
| No | 52 | 54.7 | 50 | 48.5 | | | |
| Yes | 43 | 45.3 | 53 | 51.5 | 0.759 | 0.086 | 0.384 |
| Had difficulties with relationships due to NTDs | | | | | | | |
| Yes | 51 | 53.7 | 45 | 43.7 | | | |
| No | 44 | 46.3 | 58 | 56.3 | 1.977 | 0.141 | 0.160 |
| Denied employment due NTDs | | | | | | | |
| Yes | 51 | 53.7 | 48 | 46.6 | | | |
| No | 44 | 46.3 | 55 | 53.4 | 1.415 | 0.121 | 0.234 |
| Isolated due to NTDs | | | | | | | |
| Yes | 44 | 46.3 | 36 | 35.0 | | | |
| No | 51 | 53.7 | 67 | 65.0 | 2.651 | 0.166 | 0.104 |
| Sent out of school because of NTDs | | | | | | | |
| Yes | 34 | 35.8 | 34 | 33.0 | | | |
| No | 61 | 64.2 | 69 | 67.0 | 0.169 | 0.042 | 0.681 |
| Prevented from entering a car because of NTDs | | | | | | | |
| Yes | 38 | 40.0 | 40 | 38.8 | | | |
| No | 57 | 60.0 | 63 | 61.2 | 0.028 | 0.247 | 0.867 |
| Sitting in the same car with anyone with any NTD | | | | | | | |
| Yes | 99 | 27.9 | 152 | 42.0 | | | |
| No | 161 | 45.4 | 107 | 29.6 | 22.07 | 0.113 | < 0.001 |
| Allow children to attend school where there is a known NTD case | | | | | | | |
| Yes | 80 | 22.5 | 156 | 43.1 | | | |
| No | 180 | 50.7 | 103 | 28.5 | 45.42 | 0.354 | < 0.001 |
| Employing anyone with NTD | | | | | | | |
| Yes | 53 | 14.9 | 117 | 32.3 | | | |
| No | 207 | 58.3 | 142 | 39.2 | 36.198 | 0.317 | < 0.001 |
| Continue being friends when a friend suddenly develops NTDs | | | | | | | |
| Yes | 108 | 30.4 | 156 | 43.1 | | | |
| No | 152 | 42.8 | 103 | 28.5 | 18.141 | 0.224 | < 0.001 |
| Patronizing a seller with any of the NTDs | | | | | | | |
| Yes | 79 | 22.3 | 131 | 36.2 | | | |
| No | 181 | 60.0 | 128 | 35.4 | 21.965 | 0.241 | < 0.001 |

*(Continued)*

**Table 7.** (Continued)

| Likely Stigmatization factors | Pretest | | Posttest | | (χ²) | V | P |
|---|---|---|---|---|---|---|---|
| | Freq | % | Freq | % | | | |
| Rendering assistance to anyone with obvious NTD | | | | | | | |
| Yes | 114 | 32.1 | 153 | 42.3 | | | |
| No | 146 | 41.1 | 106 | 29.3 | 12.044 | 0.181 | < 0.001 |

**Table 8.** Respondents' perception about the public artwork.

| Variable | Response | Frequency | Percentage |
|---|---|---|---|
| Seen the NTDs Public Art in the community (n = 362) | Yes | 345 | 95.3% |
| | No | 17 | 4.7% |
| Support for maintaining the Artwork in the community (n = 345) | Yes | 339 | 98.3% |
| | No | 6 | 1.7% |

the willingness to assist someone with visible NTD symptoms increased from 32.1% to 42.3% (χ² = 12.044, p < 0.001, V = 0.181).

## Respondents' perception about the public artwork

Two additional questions were added to the follow-up questionnaire, with the intention of gauging the people's perception about the public are initiative. Table 8 shows that 95.3% of the follow-up participants had encountered the Public Art murals in the community, and 98.3% of those who had seen the Artwork wanted it to be maintained in the community.

## Qualitative study findings

Table 9 outlines the characteristics of the key informants, who were leaders and healthstakeholders used in the study. There were in all 16 of them which comprised of nine males and seven females and their age raged from 31 years to 75 years. Their years of leadership in the community were between 4 and 51 years.

## Theme 1: "Awareness and knowledge"

In this theme, the community leaders expressed good awareness and knowledge of the NTDs due to the display of public art in the community.

*"I can vividly see what the NTDs are all about from the picture display" (TUC).*

*"I can now understand that the display about NTDs is not against herbal remedies to cure of the diseases. **(TH3).***

*"We always talk about the public art in the market and my members can now understand that these are curable and preventable disease" **(MUL1).***

*"I'm glad that this public art has made many to understand that NTDs are real" (CHW1)*

## Theme 2: Attitude towards NTDs

The following are some of the responses which indicate positive attitude

*"My society can now understand that these are diseases and not as a result witchcraft, and therefore welcome everyone in the society" (TL).*

**Table 9. Outline characteristics of the key informants used in the study (n = 16).**

| Code | Designation | Years of leadership in the community | Age (in Years) | Gender |
|------|-------------|--------------------------------------|----------------|--------|
| TR | Traditional Ruler | 22 years | 71 | Male |
| TUC | Town Union Chairman | 12 years | 55 | Male |
| TH1 | Traditional Herbalists 1 | 31Years | 67 | Male |
| TH2 | Traditional Herbalists 2 | 51 Years | 75 | Female |
| TH3 | Traditional Herbalists 3 | 27 years | 56 | Male |
| MUL1 | Market Union Leader 1 | 9 years | 63 | Male |
| MUL2 | Market Union Leader 2 | 6 years | 47 | Female |
| YL1 | Youth Leader 1 | 9 years | 41 | Male |
| YL2 | Youth Leader 2 | 6 years | 31 | Female |
| WL1 | Women Leader 1 | 9 years | 64 | Female |
| WL2 | Women Leader 2 | 6 years | 51 | Female |
| RL1 | Religious Leader 1 | 7 years | 44 | Female |
| RL2 | Religious Leader 2 | 10 years | 62 | Male |
| RL3 | Religious Leader 3 | 4 years | 32 | Male |
| CHW1 | Community Health Worker | 13 years | 51 | Female |
| CHW1 | Community Health Worker | 8 years | 32 | Male |

*The discrimination and fear of contracting the disease has reduced to a resonance extent"* (YL2)

*"People with NTDs can now associate freely in the society with hope of getting cure to the disease"* (WL1)

Many in the community can now believe that these diseases were not as a result of curse from gods (RL2)

**Theme 3: Practice regarding NTDs control and prevention**

The responses for practices against NTDs were generally encouraging after the public art and education to the community about NTDs. Here are some of the responses:

*This study is highly welcomed in our society and we expect it to guide our community against NTDs* (TR)

*"Many are now attending hospitals for screening for NTDs which is good news"* (CHW1)

*"Women have been very motivated by this study and we have been mobilizing people in that condition for hospital treatment"* (WL2)

*"Many of our youths have participated in training and workshops on NTDs recently and our youth association will continue to encourage active participations to clear out these diseases"* (YL1).

## Discussion

The use of public art as a health intervention represents an innovative approach for improving knowledge, attitudes, and practices regarding NTDs, particularly in local communities where misconceptions and stigma hinder effective disease control. This study sought to assess the influence of community public art in enhancing knowledge, reshaping attitudes, encouraging preventive practices, and reducing social stigmatization surrounding NTDs in Okwelle Community, Imo State, Nigeria. The findings provide strong evidence of its effectiveness in achieving these goals.

Baseline knowledge of NTDs in the study area was generally poor, with only a small proportion of respondents having prior awareness of the diseases including Onchocerciasis, Buruli ulcer and Lymphatic filariasis. This aligns with findings from a study [18] in an endemic town in Oyo State, Nigeria, where most participants had poor knowledge of Lymphatic filariasis. Similarly, Fitzpatrick et al. [19] reported widespread gaps in NTDs awareness, reinforcing the need for innovative health education strategies. Conversely, a study [11] in Nigeria reported higher awareness levels, suggesting that knowledge disparities may be influenced by sociocultural factors and previous exposure to health campaigns.

The intervention led to significant improvements in knowledge, with more respondents correctly identifying NTDs manifestations and understanding their risk. More importantly, awareness of Onchocerciasis rose from 28.5% to 39%, while knowledge of Buruli ulcer and Lymphatic filariasis also increased significantly. Improved recognition of disease symptoms and transmission patterns further supports the argument that community-based interventions play a crucial role in bridging knowledge gaps [20]. The success of this public art intervention suggests that visual and participatory education methods may be more effective than conventional health education approaches in rural communities.

Prior to the intervention, attitudes toward NTDs were characterized by misconceptions and deeply ingrained stigma. Many participants held beliefs that NTDs were hereditary, a result of witchcraft, or divine punishment. Similar attitudes have been documented in previous studies, where supernatural explanations were common barriers to seeking medical treatment [21]. However, after exposure to the public art, there was a substantial shift in perceptions. For instance, the proportion of respondents who disagreed that NTDs were a curse from God increased significantly. This aligns with findings from another study [17], which highlight that stigma and discrimination against individuals with NTDs often stem from fear and misinformation. Reducing stigma is essential for increasing healthcare-seeking behaviors and promoting social inclusion. The success of this intervention demonstrates that using culturally relevant and visually engaging methods can foster empathy, dispel myths, and encourage positive attitudinal changes in affected communities.

Baseline findings revealed poor health-seeking behaviors, with only 26% seeking medical treatment for NTD-related symptoms. This low engagement in healthcare services is consistent with prior studies that have linked poor knowledge to inadequate preventive behaviors [20]. However, post-intervention findings showed a significant increase in preventive practices, with participation in community health programs rising from 47.8% to 71.3% and active sharing of NTD-related information increasing from 37.8% to 68.0%. The improvement in preventive measures, such as personal hygiene and protective clothing use, supports earlier research indicating that targeted educational campaigns are crucial for behavior change [22]. Moreover, the increase in respondents who believed that treatment drugs should be prescribed by healthcare professionals (from 73.5% to 85.4%) suggests a growing trust in formal medical interventions, a crucial step toward effective disease control.

One of the most notable findings was the reduction in stigmatization. While stigma-related barriers to healthcare access have been widely documented [21,23], this study provides empirical evidence that public art interventions can actively counteract these social barriers. Willingness to associate with individuals affected by NTDs improved across multiple indicators, including willingness to sit in the same vehicle (27.9% to 42%) and employ affected individuals (14.9% to 32.3%). These findings highlight the potential for community-driven interventions to challenge stereotypes and promote inclusivity.

Additionally, qualitative insights from key informants reinforced this shift in perception. Participants expressed greater acceptance of individuals with NTDs, recognizing that these diseases were neither supernatural nor untreatable. This aligns with previous findings that stigma reduction is a critical component of effective disease management interventions [22]. The ability of public art to evoke emotional responses and facilitate dialogue suggests its viability as a scalable tool for addressing stigma across different NTDs-endemic regions.

The overwhelming acceptance of the public art initiative (95.3% of respondents reported encountering the murals, and 98.3% wanted them to remain) underscores its relevance as a community-driven health education strategy. Given its success, integrating public art into mainstream health communication strategies could enhance the reach and impact of NTDs awareness campaigns.

The findings from this study have important implications for other NTDs-endemic regions. Public art interventions, when designed with community participation, can be adapted to various socio-cultural contexts to enhance public health education. This aligns with global efforts to integrate creative, community-driven strategies into disease control programs. Furthermore, increasing NTDs knowledge and reducing stigma can lead to higher rates of early diagnosis and treatment adherence, ultimately supporting disease elimination goals.

## Strengths and limitations

A key strength of this study is its mixed-methods approach, combining quantitative surveys with qualitative insights from key informant interviews and focus group discussions. This allowed for a comprehensive evaluation of the intervention's impact. Additionally, the use of community-based participatory research ensured strong local engagement, which may have contributed to the effectiveness of the intervention.

However, some limitations should be acknowledged. The study was conducted within a single community, which may limit the generalizability of the findings to broader populations. Additionally, while short-term improvements in knowledge and attitudes were observed, long-term retention and behavioral changes were not assessed. Future studies should incorporate follow-up assessments to determine whether these changes are sustained over time. Potential biases in self-reported data should also be considered, as social desirability may have influenced responses. Another important limitation of this study is the exclusion of individuals under 18 years of age. While this decision was made based on ethical and methodological grounds, it also means that potential insights into the awareness and perception of NTDs among children and adolescents were not captured. Given that young people can be important targets for health education and behavioural change, future studies should consider age-appropriate methods to explore how public art influences their knowledge and attitudes toward NTDs.

## Impact on existing knowledge

This study builds on previous research demonstrating the role of culturally relevant interventions in health education. Prior studies have highlighted the effectiveness of visual storytelling and participatory methods in increasing health literacy [24]. However, few studies have specifically evaluated public art as a tool for NTDs awareness and stigma reduction. The significant improvements observed in this study suggest that public art could complement existing health education strategies, particularly in low-literacy settings where conventional approaches may be less effective.

## Conclusion and future directions

The findings from this study suggest that public art interventions can serve as a powerful, scalable strategy for enhancing NTDs awareness, improving attitudes, and reducing stigma. Compared to traditional health education methods, public art provides an engaging, culturally relevant, and visually impactful approach to disease control. Future research should explore the long-term sustainability of such interventions, assess their replicability in different socio-cultural contexts, and investigate their impact on healthcare-seeking behaviors over time. Strengthening collaborations between public health professionals, artists, and policymakers could further optimize this approach and contribute to global NTDs elimination efforts.

## Supporting information

**S1 Public Art. Public Art on NTDs in Okwele, SouthEastern, Nigeria.**
(TIF)

**S1 Dataset. Dataset of results.**
(XLSX)

## Acknowledgments

**We extend our sincere gratitude to the Community Health Extension Workers (CHEWs) and Community Health Officers (CHOs) at the Okwelle Basic Health Center for their invaluable support during this study. We also appreciate the traditional leader, religious leaders, and other community stakeholders who actively participated in the Focus Group Discussions (FGDs) and Key Informant Interviews (KIIs).**

Special thanks to the Department of Public Health, Federal University of Technology, for reviewing the study protocol and providing ethical clearance. We acknowledge the dedicated efforts of our research assistants whose contributions were instrumental in data collection and community engagement. Finally, we deeply appreciate the study participants for their willingness to share their experiences and insights, making this research possible.

## Author contributions

**Conceptualization:** Uchechukwu Madukaku Chukwuocha.

**Data curation:** Uchechukwu Madukaku Chukwuocha, Ayoola O. Bosede, Chidubem D. Osuji, Adanna N. Chukwuocha, Harriet Abugewa, Amarachi Amawuru, Adesua Okoroh, Akuchi Echefula, Precious Nwabueze, Nathan Ukwenga, Alfred Barile.

**Formal analysis:** Ayoola O. Bosede, Chidubem D. Osuji.

**Investigation:** Uchechukwu Madukaku Chukwuocha, Chidubem D. Osuji, Adanna N. Chukwuocha, Harriet Abugewa, Amarachi Amawuru, Adesua Okoroh, Akuchi Echefula, Precious Nwabueze, Nathan Ukwenga, Alfred Barile.

**Methodology:** Uchechukwu Madukaku Chukwuocha.

**Project administration:** Uchechukwu Madukaku Chukwuocha.

**Supervision:** Uchechukwu Madukaku Chukwuocha.

**Writing – original draft:** Ayoola O. Bosede, Chidubem D. Osuji.

**Writing – review & editing:** Uchechukwu Madukaku Chukwuocha, Adanna N. Chukwuocha, Harriet Abugewa, Amarachi Amawuru, Adesua Okoroh, Akuchi Echefula, Precious Nwabueze, Nathan Ukwenga, Alfred Barile.

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
