## [Decision Letter · Decision Letter 0]

PNTD-D-25-00200

Using Community Public Art to Improve Knowledge, Attitude, Practices and Mitigation of Stigmatization Regarding Neglected Tropical Diseases in a Rural Endemic Community of South Eastern Nigeria

Dear Dr. Chukwuocha,

Thank you for submitting your manuscript to PLOS Neglected Tropical Diseases. After careful consideration, we feel that it has merit but does not fully meet PLOS Neglected Tropical Diseases's publication criteria as it currently stands. Therefore, we invite you to submit a revised version of the manuscript that addresses the points raised during the review process.

Please submit your revised manuscript within 60 days May 23 2025 11:59PM. If you will need more time than this to complete your revisions, please reply to this message or contact the journal office at plosntds@plos.org. Please include the following items when submitting your revised manuscript:

We look forward to receiving your revised manuscript.

Kind regards,

Amy J Davis, Ph.D.

Academic Editor

Qu Cheng

Section Editor

Shaden Kamhawi

co-Editor-in-Chief

Paul Brindley

co-Editor-in-Chief

**Additional Editor Comments:**

This is a valuable study on public art as a health intervention, however important revisions are needed to improve clarity and organization. The reviewers have identified key areas for improvement including restructuring the methods section to clearly distinguish qualitative and quantitative approaches, strengthening the justification for the study design and location, and addressing ethical approval details. The results should emphasize effect sizes rather than solely relying on p-values, and qualitative findings need a dedicated section with a discussion of reliability.

**Journal Requirements:**

1) Please upload all main figures as separate Figure files in .tif or .eps format. For more information about how to convert and format your figure files please see our guidelines: 

2) Some material included in your submission may be copyrighted. According to PLOSu2019s copyright policy, authors who use figures or other material (e.g., graphics, clipart, maps) from another author or copyright holder must demonstrate or obtain permission to publish this material under the Creative Commons Attribution 4.0 International (CC BY 4.0) License used by PLOS journals. Please closely review the details of PLOSu2019s copyright requirements here: PLOS Licenses and Copyright. If you need to request permissions from a copyright holder, you may use PLOS's Copyright Content Permission form.

Potential Copyright Issues:

- Please confirm that you are the photographer of Plate 1, or provide written permission from the photographer to publish the photo under our CC BY 4.0 license.

3) We note that your Data Availability Statement is currently as follows: "All relevant data are within the manuscript and its Supporting Information files". Please confirm at this time whether or not your submission contains all raw data required to replicate the results of your study. Authors must share the “minimal data set” for their submission. PLOS defines the minimal data set to consist of the data required to replicate all study findings reported in the article, as well as related metadata and methods (https://journals.plos.org/plosone/s/data-availability#loc-minimal-data-set-definition).

- The points extracted from images for analysis..

**Reviewers' Comments:**

Reviewer's Responses to Questions

**Key Review Criteria Required for Acceptance?**

**Methods:**

-Are the objectives of the study clearly articulated with a clear testable hypothesis stated?

-Is the study design appropriate to address the stated objectives?

-Is the population clearly described and appropriate for the hypothesis being tested?

-Is the sample size sufficient to ensure adequate power to address the hypothesis being tested?

-Were correct statistical analysis used to support conclusions?

-Are there concerns about ethical or regulatory requirements being met?

Reviewer #1: comment has been written in the review document uploaded

Reviewer #2: Ethics: Kindly add a statement that reflects that the study was carried out in accordance with Helsinki’s declaration, also add the ethical approval institution details and the ethical approval number. This will be helpful. Very.

Study design: Would be good to mention why a mixed-methods approach was best and reference other global best practices that reflect this choice of design. Kindly add a map of the location if possible. Can you also justify any reason for the choice of Okwelle in Onuimo? Any particular reason would be interesting to mention in this scientific piece. Also, rename that section to ‘Study design/location’ – since you are describing these two components

Community engagement, stakeholder identification, and Public Art deployment: Can you split this section into two: Where do you describe the parts initially around who was engaged and all. But for the part where they already started giving feedback relevant to the study, you could move to a section under the results/findings as ‘initial findings from stakeholder engagement’

Instruments for data collection: Rename as Instruments for data collection

While the interview guide’s purpose is clear, it lacks examples of specific prompts or questions, making it hard to assess depth or relevance. Include one or two example prompts, e.g., “Prompts included questions like, ‘How has community stigma toward NTDs changed over the past year?’ and ‘What interactions have you observed between residents and the public art?’”

While ODK is mentioned as efficient, there’s no detail on how it was implemented (e.g., training for enumerators, handling of technical issues), which could affect data quality in resource-constrained settings.

Reviewer #3: The study objectives are clear. The study design addressed the stated objectives.

**Results:**

-Does the analysis presented match the analysis plan?

-Are the results clearly and completely presented?

-Are the figures (Tables, Images) of sufficient quality for clarity?

Reviewer #1: comment has been written in the review document uploaded

Reviewer #2: Results: Kindly confirm that it is scientific to have a p-value of 0.000, I think the best practice is to add 1 at the end. So, instead of 0.000, you can have 0.0001. It is also important to note that large samples always give significant values when tested using chi-square. A large sample size can indeed increase statistical power, making it easier to detect even small differences or effects as statistically significant. However, this doesn’t automatically mean the results are meaningful or practically important. With a very large sample, even trivial differences can yield significant p-values, which might mislead readers into overinterpreting the findings. I would recommend revising the manuscript to emphasize effect sizes, address multiple testing if applicable, and temper claims of importance based on p-values alone. This would strengthen the study’s credibility and avoid overhyping statistically significant but practically negligible findings.

Kindly create a separate section for qualitative findings and describe the themes that emerged from the findings and then report accordingly. Also you will need a separate section in your methods that describes the trustworthiness of your qualitative findings: Trustworthiness (credibility, transferability, dependability, and confirmability) is essential to establish the validity of qualitative findings, I do not think that the authors explicitly address how these were ensured. Strengthening these components in the methods section will enhance the study’s quality and align it with qualitative research standards. Credibility: Describe strategies used to ensure the findings reflect participants’ perspectives, such as member checking (returning findings to participants for validation) or triangulation (comparing data from interviews, FGDs, and different participant groups). Transferability: Provide thicker descriptions of the study context (e.g., characteristics of the selected LGAs or communities) to help readers assess the applicability of findings to other settings. Dependability: Detail the steps taken to ensure consistency, such as maintaining an audit trail of methodological decisions or using a standardized interview guide. Confirmability: Clarify how researcher bias was minimized, such as through peer debriefing or external review of the coding process.

Reviewer #3: The analysis was appropriate. The results were clearly presented and figures were of good quality.

**Conclusions:**

-Are the conclusions supported by the data presented?

-Are the limitations of analysis clearly described?

-Do the authors discuss how these data can be helpful to advance our understanding of the topic under study?

-Is public health relevance addressed?

Reviewer #1: yes

Reviewer #2: Great

Reviewer #3: The conclusions and limitations are appropriate. The public health relevance was addressed.

**Editorial and Data Presentation Modifications?**

Reviewer #1: comment has been written in the review document uploaded

Reviewer #2: (No Response)

Reviewer #3: 42-Abstract

48- Methodology/Principal Findings

50-51 Public art installations depicting common NTDs in the study location including, Onchocerciasis, Buruli ulcer, and Lymphatic filariasis- Were there public arts for other NTDs apart from those mentioned? If yes, mention them.

167-Study area

173-174- How did the sighted cases of NTDs confirm the prevalence of NTDs in the community?

180-Sample size

183- Please mention the previous studies where the estimated prevalence was gotten.

185-Sampling technique

192- Who were the key informants? Were they health workers, traditional rulers? Please specify their positions in the community.

197-Instrument for Data collection?

Why were open-ended questions and semi-structured questionnaires not used in this study?

279-Results

318-Table 2- Avoiding being biting by blackflies. Sentence needs revision and was duplicated. Please correct.

359-364-What is the meaning of the abbreviations of the respondents’ names?

398-404-What is the meaning of the abbreviations of the respondents’ names?

**Summary and General Comments:**

Reviewer #1: comment has been written in the review document uploaded

Reviewer #2: Introduction: Kindly reword the last paragraph of your introduction to the past tense. Check through and correct this

Ethics: Kindly add a statement that reflects that the study was carried out in accordance with Helsinki’s declaration, also add the ethical approval institution details and the ethical approval number. This will be helpful. Very.

Study design: Would be good to mention why a mixed-methods approach was best and reference other global best practices that reflect this choice of design. Kindly add a map of the location if possible. Can you also justify any reason for the choice of Okwelle in Onuimo? Any particular reason would be interesting to mention in this scientific piece. Also, rename that section to ‘Study design/location’ – since you are describing these two components

Community engagement, stakeholder identification, and Public Art deployment: Can you split this section into two: Where do you describe the parts initially around who was engaged and all. But for the part where they already started giving feedback relevant to the study, you could move to a section under the results/findings as ‘initial findings from stakeholder engagement’

Instruments for data collection: Rename as Instruments for data collection

While the interview guide’s purpose is clear, it lacks examples of specific prompts or questions, making it hard to assess depth or relevance. Include one or two example prompts, e.g., “Prompts included questions like, ‘How has community stigma toward NTDs changed over the past year?’ and ‘What interactions have you observed between residents and the public art?’”

While ODK is mentioned as efficient, there’s no detail on how it was implemented (e.g., training for enumerators, handling of technical issues), which could affect data quality in resource-constrained settings.

Results: Kindly confirm that it is scientific to have a p-value of 0.000, I think the best practice is to add 1 at the end. So, instead of 0.000, you can have 0.0001. It is also important to note that large samples always give significant values when tested using chi-square. A large sample size can indeed increase statistical power, making it easier to detect even small differences or effects as statistically significant. However, this doesn’t automatically mean the results are meaningful or practically important. With a very large sample, even trivial differences can yield significant p-values, which might mislead readers into overinterpreting the findings. I would recommend revising the manuscript to emphasize effect sizes, address multiple testing if applicable, and temper claims of importance based on p-values alone. This would strengthen the study’s credibility and avoid overhyping statistically significant but practically negligible findings.

Kindly create a separate section for qualitative findings and describe the themes that emerged from the findings and then report accordingly. Also you will need a separate section in your methods that describes the trustworthiness of your qualitative findings: Trustworthiness (credibility, transferability, dependability, and confirmability) is essential to establish the validity of qualitative findings, I do not think that the authors explicitly address how these were ensured. Strengthening these components in the methods section will enhance the study’s quality and align it with qualitative research standards. Credibility: Describe strategies used to ensure the findings reflect participants’ perspectives, such as member checking (returning findings to participants for validation) or triangulation (comparing data from interviews, FGDs, and different participant groups). Transferability: Provide thicker descriptions of the study context (e.g., characteristics of the selected LGAs or communities) to help readers assess the applicability of findings to other settings. Dependability: Detail the steps taken to ensure consistency, such as maintaining an audit trail of methodological decisions or using a standardized interview guide. Confirmability: Clarify how researcher bias was minimized, such as through peer debriefing or external review of the coding process.

Reviewer #3: The research is a novel one and has contributed to the knowledge needed in the control of Neglected Tropical Diseases.

PLOS authors have the option to publish the peer review history of their article (what does this mean? ). If published, this will include your full peer review and any attached files.

**Do you want your identity to be public for this peer review?** For information about this choice, including consent withdrawal, please see our Privacy Policy .

Reviewer #1: **Yes: ** Dr. Taiwo Mofadeke Jaiyeola

Reviewer #2: No

Reviewer #3: No

**Figure resubmission:**
---

## [Decision Letter · Decision Letter 1]

PNTD-D-25-00200R1Assessment of the Effectiveness of Public Art in improving Knowledge, Attitude, Practices and Mitigation of Stigmatization regarding Neglected Tropical Diseases in South Eastern, NigeriaPLOS Neglected Tropical Diseases Dear Dr. Chukwuocha, Thank you for submitting your manuscript to PLOS Neglected Tropical Diseases. After careful consideration, we feel that it has merit but does not fully meet PLOS Neglected Tropical Diseases's publication criteria as it currently stands. Therefore, we invite you to submit a revised version of the manuscript that addresses the points raised during the review process.

Please submit your revised manuscript within 30 days Jun 17 2025 11:59PM. If you will need more time than this to complete your revisions, please reply to this message or contact the journal office at plosntds@plos.org. Please include the following items when submitting your revised manuscript: * A rebuttal letter that responds to each point raised by the editor and reviewer(s). You should upload this letter as a separate file labeled 'Response to Reviewers '. This file does not need to include responses to any formatting updates and technical items listed in the 'Journal Requirements' section below.* A marked-up copy of your manuscript that highlights changes made to the original version. You should upload this as a separate file labeled 'Revised Manuscript with Track Changes '.* An unmarked version of your revised paper without tracked changes. You should upload this as a separate file labeled 'Manuscript '. If you would like to make changes to your financial disclosure, competing interests statement, or data availability statement, please make these updates within the submission form at the time of resubmission. Guidelines for resubmitting your figure files are available below the reviewer comments at the end of this letter.

We look forward to receiving your revised manuscript.

Kind regards,

Amy J Davis, Ph.D.

Academic Editor

PLOS Neglected Tropical Diseases Qu ChengSection EditorPLOS Neglected Tropical Diseases

Shaden Kamhawi

co-Editor-in-Chief

Paul Brindley

co-Editor-in-Chief

**Additional Editor Comments :** The authors have done a really nice job responding to reviewer comments. Reviewers 1 & 3 have some additional comments to help improve the clarity of the manuscript. Please place close attention to those suggestions in your revision. **Journal Requirements:** 1) The following file is currently uploaded as file type 'Other', which is not viewable by the reviewers: Plate 1 Public Art on Neglected Tropical Diseases.tif . If you wish it to be included in review, please change the file type to 'Supporting Information' and include a legend in the manuscript. Please also ensure to remove the logo or replace it as we are not permitted to publish this under our CC-BY 4.0 license, even with permission. **Comments to the Authors:****Please note that one of the reviews is uploaded as an attachment.** **Reviewers' comments:** Reviewer's Responses to Questions

**Key Review Criteria Required for Acceptance?**

**Methods**

-Are the objectives of the study clearly articulated with a clear testable hypothesis stated?

-Is the study design appropriate to address the stated objectives?

-Is the population clearly described and appropriate for the hypothesis being tested?

-Is the sample size sufficient to ensure adequate power to address the hypothesis being tested?

-Were correct statistical analysis used to support conclusions?

-Are there concerns about ethical or regulatory requirements being met?

Reviewer #1: (No Response)

Reviewer #2: Authors have now addressed the comments raised satisfactorily. This strengthens the work and I can recommend acceptance of the paper to be published!

Reviewer #3: The objectives were clearly stated and the study design was appropriate.

**Results**

-Does the analysis presented match the analysis plan?

-Are the results clearly and completely presented?

-Are the figures (Tables, Images) of sufficient quality for clarity?

Reviewer #1: yes but with minimal corrections

Reviewer #2: Authors have now addressed the comments raised satisfactorily. This strengthens the work and I can recommend acceptance of the paper to be published!

Reviewer #3: The results were clearly presented and well analysed.

**Conclusions**

-Are the conclusions supported by the data presented?

-Are the limitations of analysis clearly described?

-Do the authors discuss how these data can be helpful to advance our understanding of the topic under study?

-Is public health relevance addressed?

Reviewer #1: yes

Reviewer #2: Authors have now addressed the comments raised satisfactorily. This strengthens the work and I can recommend acceptance of the paper to be published!

Reviewer #3: The conclusions are supported by the data presented.

**Editorial and Data Presentation Modifications?**

Reviewer #1: yes, edit tables to improve the outlook

Reviewer #2: Authors have now addressed the comments raised satisfactorily. This strengthens the work and I can recommend acceptance of the paper to be published!

Reviewer #3: 170-Sampling technique

180- Who were the key informants? Were they health workers, traditional rulers? Please specify their positions in the community.

202-Instrument for Data collection Quantitative data?-Review sentence

Why were open-ended questions and semi-structured questionnaires not used in this study?

380-Results

427-Table 2- Avoiding being biting by blackflies. Sentence needs revision and was duplicated. Please correct.

555 Qualitative study findings

What do you mean by leaders/ stakeholders used in the study? Please provide clearer explanation of their roles in the study. At what point were they used and what were they used for?

**Summary and General Comments**

Reviewer #1: A great improvement from the last review but there are still some corrections to be made

Reviewer #2: Authors have now addressed the comments raised satisfactorily. This strengthens the work and I can recommend acceptance of the paper to be published!

Reviewer #3: General comments: The study is an interesting and novel one and has added to scientific knowledge. However, it would have been more robust if participants lesser than 18 years were included in the study to demonstrate if children would gain knowledge from viewing these art works. I am curious to know why children were exempted from this study though.

PLOS authors have the option to publish the peer review history of their article (what does this mean? ). If published, this will include your full peer review and any attached files.

**Do you want your identity to be public for this peer review?** For information about this choice, including consent withdrawal, please see our Privacy Policy .

Reviewer #1: **Yes: ** Taiwo Mofadeke Jaiyeola

Reviewer #2: No

Reviewer #3: No

---

## [Editor Report · Decision Letter 2]

PNTD-D-25-00200R2Assessment of the Effectiveness of Public Art in improving Knowledge, Attitude, Practices and Mitigation of Stigmatization regarding Neglected Tropical Diseases in South Eastern, NigeriaPLOS Neglected Tropical DiseasesDear Dr. Chukwuocha, Thank you for submitting your manuscript to PLOS Neglected Tropical Diseases. After careful consideration, we feel that it has merit but does not fully meet PLOS Neglected Tropical Diseases's publication criteria as it currently stands. Therefore, we invite you to submit a revised version of the manuscript that addresses the points raised during the review process. Please submit your revised manuscript within 30 days Jun 26 2025 11:59PM. If you will need more time than this to complete your revisions, please reply to this message or contact the journal office at plosntds@plos.org. Please include the following items when submitting your revised manuscript: * A rebuttal letter that responds to each point raised by the editor and reviewer(s). You should upload this letter as a separate file labeled 'Response to Reviewers '. This file does not need to include responses to any formatting updates and technical items listed in the 'Journal Requirements' section below. * A marked-up copy of your manuscript that highlights changes made to the original version. You should upload this as a separate file labeled 'Revised Manuscript with Track Changes '. * An unmarked version of your revised paper without tracked changes. You should upload this as a separate file labeled 'Manuscript '. If you would like to make changes to your financial disclosure, competing interests statement, or data availability statement, please make these updates within the submission form at the time of resubmission. Guidelines for resubmitting your figure files are available below the reviewer comments at the end of this letter. We look forward to receiving your revised manuscript. Kind regards, Amy J Davis, Ph.D.Academic EditorPLOS Neglected Tropical Diseases Qu ChengSection EditorPLOS Neglected Tropical Diseases

Shaden Kamhawi

co-Editor-in-Chief

Paul Brindley

co-Editor-in-Chief

**Additional Editor Comments:** A previous reviewer had questioned why individuals under 18 years of age were not included in the study. The authors have responded to that only in the responses to reviewer comments but not addressed it within the manuscript itself. Think it’s important to put the rationale for that decision in the methods section, and to add in the discussion section possible implications for not including those under 18 and whether or not the authors think it would be worthwhile to include them in future studies.**Reviewers' comments:** **Figure resubmission:** While revising your submission, please upload your figure files to the Preflight Analysis and Conversion Engine (PACE) digital diagnostic tool, https://pacev2.apexcovantage.com/. PACE helps ensure that figures meet PLOS requirements. To use PACE, you must first register as a user. Registration is free. Then, login and navigate to the UPLOAD tab, where you will find detailed instructions on how to use the tool. If you encounter any issues or have any questions when using PACE, please email PLOS at figures@plos.org. Please note that Supporting Information files do not need this step. If there are other versions of figure files still present in your submission file inventory at resubmission, please replace them with the PACE-processed versions.  **Reproducibility:** To enhance the reproducibility of your results, we recommend that authors of applicable studies deposit laboratory protocols in protocols.io, where a protocol can be assigned its own identifier (DOI) such that it can be cited independently in the future. Additionally, PLOS ONE offers an option to publish peer-reviewed clinical study protocols. Read more information on sharing protocols at https://plos.org/protocols?utm_medium=editorial-email&utm_source=authorletters&utm_campaign=protocols

---

## [Editor Report · Decision Letter 3]

PNTD-D-25-00200R3Assessment of the Effectiveness of Public Art in improving Knowledge, Attitude, Practices and Mitigation of Stigmatization regarding Neglected Tropical Diseases in South Eastern, NigeriaPLOS Neglected Tropical Diseases Dear Dr. Chukwuocha,

Thank you for submitting your manuscript to PLOS Neglected Tropical Diseases. After careful consideration, we feel that it has merit but does not fully meet PLOS Neglected Tropical Diseases's publication criteria as it currently stands. Please see '**Additional Editor Comments'**  section for remaining comment to address. Therefore, we invite you to submit a revised version of the manuscript that addresses the points raised during the review process.

Please note that the "Additional Editor Comment" must be addressed before we can proceed with processing your submission.

Please submit your revised manuscript within 30 days Jul 10 2025 11:59PM. If you will need more time than this to complete your revisions, please reply to this message or contact the journal office at plosntds@plos.org. Please include the following items when submitting your revised manuscript:

* A rebuttal letter that responds to each point raised by the editor and reviewer(s). You should upload this letter as a separate file labeled 'Response to Reviewers '. This file does not need to include responses to any formatting updates and technical items listed in the 'Journal Requirements' section below.* A marked-up copy of your manuscript that highlights changes made to the original version. You should upload this as a separate file labeled 'Revised Manuscript with Track Changes '.* An unmarked version of your revised paper without tracked changes. You should upload this as a separate file labeled 'Manuscript '. If you would like to make changes to your financial disclosure, competing interests statement, or data availability statement, please make these updates within the submission form at the time of resubmission. Guidelines for resubmitting your figure files are available below the reviewer comments at the end of this letter. We look forward to receiving your revised manuscript. Kind regards, Amy J Davis, Ph.D.Academic EditorPLOS Neglected Tropical Diseases Qu ChengSection EditorPLOS Neglected Tropical Diseases

Shaden Kamhawi

co-Editor-in-Chief

Paul Brindley

co-Editor-in-Chief

**Additional Editor Comments :** A previous reviewer had questioned why individuals under 18 years of age were not included in the study. The authors have responded to that only in the responses to reviewer comments but not addressed it within the manuscript itself. Think it’s important to put the rationale for that decision in the methods section, and to add in the discussion section possible implications for not including those under 18 and whether or not the authors think it would be worthwhile to include them in future studies. **Journal Requirements:**

1) Please include the authors' affiliations in the online submission form. Please ensure that the affiliations of the authors listed on the manuscript title page do exactly match with the affiliations provided in the online submission form.

NOTE: Affiliations should include a department (if applicable), an institution, a city, and a country.

---

## [Editor Report · Decision Letter 4]

Dear Dr Chukwuocha,

We are pleased to inform you that your manuscript 'Assessment of the Effectiveness of Public Art in improving Knowledge, Attitude, Practices and Mitigation of Stigmatization regarding Neglected Tropical Diseases in South Eastern, Nigeria' has been provisionally accepted for publication in PLOS Neglected Tropical Diseases.

Best regards,

Amy J Davis, Ph.D.

Academic Editor

Qu Cheng

Section Editor

Shaden Kamhawi

co-Editor-in-Chief

Paul Brindley

co-Editor-in-Chief

---

## [Editor Report · Acceptance letter]

Dear Dr Chukwuocha,

We are delighted to inform you that your manuscript, "Assessment of the Effectiveness of Public Art in improving Knowledge, Attitude, Practices and Mitigation of Stigmatization regarding Neglected Tropical Diseases in South Eastern, Nigeria," has been formally accepted for publication in PLOS Neglected Tropical Diseases.

Best regards,

Shaden Kamhawi

co-Editor-in-Chief

Paul Brindley

co-Editor-in-Chief
